# Dosage Form Modification, a Simulation Activity between Nursing and Pharmacy Students

**DOI:** 10.3390/pharmacy10060141

**Published:** 2022-10-27

**Authors:** Chiao Xin Lim, Karen Livesay, Julie Stevens, Vivek Nooney, Katherine Baverstock, Nichole Orwin, Ieva Stupans

**Affiliations:** 1Pharmacy, School of Health and Biomedical Sciences, RMIT University, Melbourne, VIC 3000, Australia; 2Nursing, School of Health and Biomedical Sciences, RMIT University, Melbourne, VIC 3000, Australia; 3Clinical and Health Sciences, University of South Australia, Adelaide, SA 5000, Australia; 4Adelaide Medical School, Faculty of Health and Medical Sciences, University of Adelaide, Adelaide, SA 5000, Australia

**Keywords:** interprofessional education, pharmacy, nursing, simulation training, dosage form modification

## Abstract

**Background:** The aim of this exploratory pilot study was to evaluate student perceptions of a simulation activity involving undergraduate nursing and pharmacy students. The key question was “how do nursing and pharmacy students respond in an immersive collaborative simulation activity which involves medication dosage form modification?” **Methods:** One hundred nursing students participated in a simulated exercise where patients required medications for which there were barriers to administration. Fourteen pharmacy students were also present in the simulated health environment, observing the work of the nursing students and being available to provide advice regarding medication administration to the nursing students. A mixed methods approach was employed for this exploratory pilot study, where both nursing and pharmacy students were invited to complete a survey regarding the experience at the end of the simulation exercise and pharmacy students completed a reflection. Both surveys and reflections were analyzed. **Results:** Survey results indicated very high satisfaction regarding the exercise for both pharmacy and nursing students. Analysis of pharmacy student reflections also indicated apprehension regarding their preparedness to contribute to the exercise, enjoyment in participation, their understanding of the value of collaboration between the two groups of students, and also recognition of their need to be more prepared for such situations. **Conclusion:** This study assessed student perceptions and did not formally evaluate learning outcomes. The interprofessional immersive simulated learning opportunity was viewed as valuable by both nursing and pharmacy students. The immersive simulation provided teaching staff with the opportunity to develop a new approach for the teaching of dosage form modification to both nursing and pharmacy students in an interprofessional setting.

## 1. Introduction

The use of simulation across nursing education has grown rapidly, particularly for the practice of technical skills, but also for communication, critical thinking, problem-solving and decision making [1,2]. Realistic clinical scenarios, requiring the administration of medications provide students with the opportunity to apply theoretical knowledge in a practical situation. In Australia, the nursing curriculum uses a number of different teaching and learning approaches. Considerations such as the best way to administer a medication, may be addressed as theory or simulated practice. Depending on placement sites experienced during their degree, nursing students may or may not have the opportunity to observe the capabilities of pharmacists in the provision of advice around medication administration. Australian nursing curricula prepare students to meet the Registered Nurse (RN) standards for practice [3]. Within the RN standards the concepts of critical thinking, compliance with legislative requirements, and participating in collaborative practice are all relevant to medication administration exercises.

Although hospital placements are mandatory for pharmacy students in Australia and New Zealand [4], accessing hospital experiences for pharmacy students has become increasingly challenging. Pharmacy students may have some ward exposure, but are not necessarily able to observe the work of nursing staff with respect to medication administration. 

The roles of pharmacists in Australia are also currently undergoing expansion to include pharmacists located within a range of clinical areas representing differing acuity across a continuum from aged care facilities to emergency departments [5]. The importance of collaboration between nursing staff and pharmacists regarding dose administration and potential dose form modification cannot be underestimated, for example within aged care facilities [6]. For residents in aged care facilities, the oral route is the most preferred and convenient for medication administration, however, co-morbidities, age-related physiological changes, and polypharmacy may necessitate dosage form modification [7]. A recent study in aged care settings in Australia has indicated that 12.5% of dosage modification instances were inappropriate and commonly associated with suboptimal methods of medication preparation [8].

Patients in acute care medical and surgical wards receiving nutrition through enteral feeding tubes also may require dose form modification, with modification primarily undertaken at the bedside, by nursing staff [9,10]. Inappropriate dosage modification is a significant issue. Data from a study in an Australian hospital suggests that only 45% of medications are prepared safely [9]. Furthermore, issues such as time constraints, the complexity of medication administration processes, inefficient information flow, and communication among healthcare professionals, as well as nurses’ knowledge and training have been identified in aged care settings [11], and could also be considered as issues in hospitals. 

Collaboration between pharmacists and nurses regarding dosage form modification in a range of different patients in a number of settings has the potential to improve practice. Immersive simulation is an appropriate approach to address this in both nursing and pharmacy student curricula [12,13]. Although not addressing dosage form modification, a recent study has found that simulation improved nursing medication administration performance in practice [12]. To the authors’ knowledge, this paper is the first report on the use of immersive simulation between pharmacy and nursing students addressing dosage form modification as a learning opportunity for both groups of students.

## 2. Methods 

In this simulation, students were immersed in clinical scenarios based in a medical/surgical ward. Simulation exercises were designed with face validity to replicate authentic situations through which student-centered interactive problem solving occurred without risks to patients. The learning design incorporated facilitator interaction to help students reflect on the simulation experience. In these scenarios, the simulation used a ‘pause and discuss’ interaction between the facilitator and students as well as a structured debrief at the end of the session.

Students worked through four cases in the two-hour session in an immersive simulated hospital setting. For each case the students spent approximately 25–30 min working through each scenario, followed by a discussion led by a nursing instructor. There were one or two pharmacy students and 11–12 nursing students in each session. All four cases were worked through by all students. In addition to the nursing instructors, a pharmacy academic was also present at these sessions.

One of the scenarios used for nursing and pharmacy students is shown in Table 1. The second case involves evaluating appropriate dosage form modification for a patient who is unable to swallow. The other two cases focused on the compatibilities of IV medication administration.

For second year undergraduate nursing students this exercise was one of six such simulation exercises in this subject. This exercise was in the second therapeutics subject in the nursing program of study, thus nursing students were well prepared. Pharmacy students were invited to participate in the exercise as one of the learning opportunities in a subject in the final year of their four-year undergraduate program. Pharmacy students who participated were required to complete a reflective task. Students who did not participate were given an alternate written assessment.

A mixed methods approach was employed for this exploratory pilot study, where both pharmacy and nursing students completed a survey post-simulation exercises and pharmacy students completed a reflection. Quantitative evaluation data were collected from both nursing and pharmacy students, who gave permission to have their data included in the analysis, using a paper-based survey which was distributed during the simulation session. The survey was developed by the authors and focused on students’ perceptions of working together to solve patient medication problems and their perception of an interprofessional exercise. Our focus was to establish students’ enthusiasm and engagement, both key motivators for learning. There was also a free text question enabling other comments to be made by students. All survey questions utilized a 1 to 5-point Likert scale (1 = strongly disagree, 2 = disagree, 3 = neutral, 4 = agree, 5 = strongly agree). Face validity was established through reviews by both nursing and pharmacy academic staff. The data were analyzed using Excel (Microsoft^®^ 365).

The reflective accounts of pharmacy students who gave permission for analysis as part of the study were analyzed through inductive content analysis [14]. Two authors explored the interview text independently (I.S. & V.N.) by reading the text several times to obtain the overall meaning and identify noteworthy categories. After discussion consensus was reached and the categories were coded. 

Ethics permission was provided by the University Human Research Ethics Committee, approval number 25107. The project was developed and is reported following SQUIRE-EDU guidelines [15].

## 3. Results

One -hundred nursing students who participated completed the survey of whom 97 completed all survey questions. Of the 14 pharmacy students who participated in the exercise, 12 participated in the survey and nine gave permission to have their reflective accounts analyzed. Results of the survey are shown in Table 2.

Sixteen nursing students and eight pharmacy students provided open-ended responses. Given that there were only limited number of free text comments which were short, the responses have not been thematically analyzed. Three quotes from nursing students and four quotes from pharmacy students, which represented the students’ thoughts immediately after the completion of the session, are shown as follows:
“Very helpful to get a different perspective and interact with other health care professionals.”(Nursing student)
“Would be more accessible to have more than one pharmacist- unable to access pharmacist when they are with other groups.”(Nursing student)
“It was interesting seeing how a pharmacy student were (sic) able to assist us with our medications when needed.”(Nursing student)
“Overall great experience that allows you to interact with other disciplines in a portrayal of real life hospital. Also allows pharmacy students to take into account such things as enteral feeding and crushing medications which we do not get to experience or learn much about in class or community pharmacy.”(Pharmacy student)
“Very interesting workshop. We need more of these workshops to get more experience in dealing with healthcare staff in hospital settings”(pharmacy student)
“I learnt many things from this experience. Definitely helped practice interprofessional communication”(pharmacy student)
“Brilliant, so important for all professions.”(Pharmacy student)

The following themes were identified from the analysis of the pharmacy student reflections: (1) apprehension regarding their preparedness to contribute to the exercise and recognition of their need to be more prepared for such situations, (2) enjoyment in participation, (3) their understanding of the value of collaboration between the two groups of students. 

Pharmacy student participants were apprehensive prior to the exercise:
“I was feeling very nervous and stressed prior to the workshop as I was worried about not being able to answer questions asked by nursing students. Thus, feeling useless. “(Student 2)

and
“I could have understood the cases a bit better before coming so I could give more comprehensive answers regarding interactions and medicine changes.”(Student 3)

Collaboration between the participants was also observed:
“When the workshop started, all students acted professionally, being friendly and patient with one another, ensuring a very collaborative and professional environment.”(Student 1)

and
“Sharing information between students expands our knowledge and increases our appreciation and awareness of the capabilities of each healthcare professional.”(Student 2)

Lastly students commented that they had enjoyed the activity and recognized it as an important learning experience:
“Experience during the workshop was very enjoyable, and I had a great time.”(Student 9)
“It would be fantastic to experience these types of simulation classes again.”(Student 8)

## 4. Discussion

Pharmacy students indicated that they felt prepared for the experience in the survey and yet reflected recognition of the need for additional preparation as well. This apparent contradiction may have several explanations. Students were theoretically prepared by having access to the scenarios in advance of the simulation to prepare and research. This is in keeping with the Standards of Best Practice for Prebriefing [16]. However, they were not orientated to the simulated health environment until the day of the simulation. On arrival, they were briefed regarding the educational experience and ground rules for simulation [16]. Therefore, cognitive load and consequent anxiety may have been reduced with the review of salient features of the case scenarios and an understanding of the process and expectations for each scenario in advance of the day of simulation. Another possible explanation is that pharmacy students were asked to provide advice and guidance around medication administration to nursing students, which is a higher level of skill requiring a deep understanding of the topic. This may also contribute to the apprehension observed in pharmacy students’ reflections.

Both nursing and pharmacy students appreciated the opportunity to work with students from another health discipline and recognized the importance of interprofessional collaboration to improve patient care. The desire for more interprofessional education activities expressed by both groups of students is in agreement with a recent systematic review conducted by Grimes and Guinan [17], where the demand for more interprofessional education experiences was highlighted in the student feedback. The positive student satisfaction shown in this study could be attributed to the immersive simulation delivery, as there is evidence that simulation-based interprofessional activity is an effective method to promote actual learning development [17]. 

A limitation of this study is the small number of pharmacy students who participated in this exploratory pilot study. Additionally, only students’ perceptions were evaluated, which is the base level of Kirkpatrick’s levels of training evaluation [18] and is deemed as low-level evidence because of the risk of bias. Future studies could include a baseline assessment and a post-exercise assessment to evaluate knowledge acquisition which corresponds to level two of Kirkpatrick’s model, to ascertain if the simulation exercise improved the knowledge and skills of medication administration and dosage form administration in nursing and pharmacy students.

## 5. Conclusions

The aim of this exploratory pilot study was to explore nursing and pharmacy students’ experiences of participating in an immersive interprofessional simulation for which there were complexities with regard to administration of medications. The Likert survey, the sample free text comments, and the interpreted overarching themes, apprehension and recognition of the need to be more prepared, the importance of collaboration, and the enjoyment of participation provide a basis to discuss the overwhelmingly positive student perceptions regarding this activity. This exploratory pilot emphasizes the importance of these opportunities in the health profession curriculum. 

## Figures and Tables

**Table 1 pharmacy-10-00141-t001:** Example scenario used in a simulation exercise.

Mr. Tom Schaeffer is a 60-year-old male, with a lifelong history of smoking. He presented 1 week ago with a large squamous cell carcinoma of the lower lip. He underwent surgery the next day including wide, local excision and complex lip reconstruction. Five days later, he was noted to be meeting only half of his nutritional requirements as calculated by the dietician and was prescribed enteral feeding via nasogastric tube. Mr Schaeffer’s medical conditions include gastro-oesophageal reflux disease, depression, hypertension and whilst he has weaned almost completely off opioids, he is still experiencing mild post-operative pain at night. He has normal renal function. His medications include: Nexium® (esomeprazole) 20 mg tablets—1 tablet morning Pristiq® (desvenlafaxine) 50 mg tablets—1 tablet morning Exforge 5/80® (amlodipine 5 mg/valsartan 80 mg) tablets—1 tablet morning Oxynorm® (oxycodone) 5 mg capsules—1 capsule three times daily when necessary (he has weaned off post-op opioids such that he now only requires one capsule before bed) The nursing staff contact you (the pharmacist) for advice on how to administer Mr Schaeffer’s medicines via enteral feeding tube. Provide comprehensive administration instructions for each medication, making reference to tube size/type and include any warnings you consider important to convey to the nursing staff about crushing of medicines as appropriate, e.g. teratogenic potential. If it is necessary to change the medication, describe the suggestions you would make to the doctor, including drug name, dose and frequency and include instructions on how the new medication would be administered via enteral feeding tube.

**Table 2 pharmacy-10-00141-t002:** Post-simulation survey results.

Question(s)	Strongly Agree	Agree	Neither Agree Nor Disagree	Disagree	Strongly Disagree
*Nursing students (n = 100)*					
I felt comfortable asking the **pharmacy** students for advice	64	33	1	0	0
It was interesting to see the skills of **pharmacy** in dose administration	66	30	2	0	0
I can see how patient care can be improved if health professions work together	80	18	2	0	0
I would like more opportunities to work with students from other disciplines on campus	69	26	3	0	0
It was a great experience to be able to work with students from another discipline	74	22	1	0	0
*Pharmacy students (n = 12)*					
I felt comfortable providing advice to the **nursing** students	8	4	0	0	0
I was adequately prepared to be able to **provide advice**	7	5	0	0	0
It was interesting to see the skills of **nursing** in dose administration	11	1	0	0	0
I can see how patient care can be improved if health professions work together	11	1	0	0	0
I would like more opportunities to work with students from other disciplines on campus	12	0	0	0	0
It was a great experience to be able to work with students from another discipline	12	0	0	0	0

## Data Availability

Not applicable.

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
