# Peer review of "Dosage Form Modification, a Simulation Activity between Nursing and Pharmacy Students"

_pharmacy, 2022, doi:10.3390/pharmacy10060141_

Round 1
Reviewer 1 Report
The communication is about a simulation conducted between nursing and pharmacy students in the theme of dosage form modification. I found this topic really interesting. However, in my opinion, some points could be improved in the methods and discussion:
1. Authors say they presented 4 cases, but only one is in the paper. Were the other 3 cases about the same dosage modification? I suggest to write down at least a stract of the other cases;
2. Authors say that the cases were presented and discussed for 2 hours. Each case had half hour to be discussed?
3. How was the proportion of nursing to pharmacy students in the cases? All of them have discussed the 4 cases?
4. Nursing students were from the 2nd year course. Considering their academic knowledge, they were already prepared for such a discussion - thinking about dosage form technology?
I suggest to explain the codes used from the lines 89 - 92 like "mane", "tds", "prn". Even been able to read in english, depending on the specific area, the codes are not clear;
How was the real scenarium: students were in a simulated hospital ou in a college regular classroom or auditorium?
It would be interested to know what kind of questions were asked from the nursing students and how correct were the answers from the pharmacy students.
I think that it would be useful to discuss more about how prepared the health service is to conduct dosage form modifications. The realities vary a lot so, mentioning hospital structure as well as human resources to do such a task would be interesting.
Author Response
We would like to thank the reviewer for their comments. Your feedback is incredibly useful for us to revise our manuscript and improve the quality of our work.
The communication is about a simulation conducted between nursing and pharmacy students in the theme of dosage form modification. I found this topic really interesting. However, in my opinion, some points could be improved in the methods and discussion:
- Authors say they presented 4 cases, but only one is in the paper. Were the other 3 cases about the same dosage modification? I suggest to write down at least abstract of the other cases;
Reply: The second case involves evaluating appropriate dosage form administration to a patient who is unable to swallow. The other two cases focus on compatibility of IV medication administration. These details have been added.
- Authors say that the cases were presented and discussed for 2 hours. Each case had half hour to be discussed?
Reply: The students spent approximately 25-30 minutes on each case, and there was a discussion led by a nursing instructor by the end of each case.
- How was the proportion of nursing to pharmacy students in the cases? All of them have discussed the 4 cases?
Reply: There were 1-2 pharmacy students and 11-12 nursing students in each of the simulations. All 4 cases were worked through by all students.
- Nursing students were from the 2nd year course. Considering their academic knowledge, they were already prepared for such a discussion - thinking about dosage form technology?
Reply: As this is the second therapeutics course that nursing students are completing, they were well prepared for this exercise.
- I suggest to explain the codes used from the lines 89 - 92 like "mane", "tds", "prn". Even been able to read in english, depending on the specific area, the codes are not clear;
Reply: These have been revised to English to improve readability (Table 1).
- How was the real scenarium: students were in a simulated hospital out in a college regular classroom or auditorium?
Reply: This activity was carried out in a simulated hospital.
- It would be interested to know what kind of questions were asked from the nursing students and how correct were the answers from the pharmacy students.
Reply: We thank reviewer 1 for their comment. As the simulations were not recorded, we were unable to provide further information on the questions and answers from students. Informally, they mainly concerned sources of information and how to interpret information from the clinical resources.
- I think that it would be useful to discuss more about how prepared the health service is to conduct dosage form modifications. The realities vary a lot so, mentioning hospital structure as well as human resources to do such a task would be interesting.
Reply: Little detail regarding the specific requests of the reviewer is available. A recent study in aged care settings in Australia has indicated that 12.5% of dosage modification instances were inappropriate and commonly associated with suboptimal methods of medication preparation. We have added further details in the introduction to highlight the paucity of the research in this space.
Reviewer 2 Report
This manuscript describes an evaluation of an interprofessional simulation activity between nursing and pharmacy students within the context of a medication dosage form modification. The general topic of interprofessional education is important and of interest, however, there is also a large (and growing) body of literature in this area in this area describing pharmacy and nursing collaborations. It is unclear how this study contributes to the existing literature base and there are also major concerns regarding the methods used. Specific comments are provided:
Introduction
1) More information needs to be provided regarding the large body of literature addressing interprofessional education efforts between nursing and pharmacy students, including current gaps that this study is intended to address. Why is oral dosage form modification important to study specifically rather than more general medication activities and responsibilities between nurses and pharmacists? What is not transferable from existing literature that might differ for this specific activity?
Methods
2) There is no study design identified.
3) Line 73: How was face validity established? It is not enough to say it was established.
4) Consider placing the case in a table instead of in-text.
5) Lines 109-111: Why did the authors choose to create their own survey rather than use an existing (and potentially validated survey) that measured the same outcomes? Why was it important to ask questions about if students felt the activity was "interesting"? What was the utility of the item asking if this was a "great experience"?
6) Lines 118-119: Do the authors mean that the responses were coded using the identified categories? What was the outcome of this analysis (i.e. what were done with the codes)?
Results
7) Lines 133-134: It would be helpful to identify the number of students who provided open-ended responses.
8) Lines 135-149: Just listing quotes is not necessarily meaningful without some additional context to them. For example, you identify themes and associated quotes (which is a standard way to present this data), but these quotes simply stand alone with no explanation.
9) Themes are presented, but there was no description in the Methods of how you got from categories and codes to themes. If themes were developed, please discuss any qualitative validation steps taken.
Discussion
10) Line 186: It was not clear in the Methods that teaching was part of the simulation activity. Please provide more detail in the Methods about the responsibilities of each profession during the simulation.
11) Lines 190-198: If this is a major takeaway from the study, it does not add to the existing literature base and is not specific to the oral dosage form modification activity that supposedly is the unique contributor of this study.
12) Lines 199-206: Other interprofessional education studies have addressed these limitations or did not have these limitations present, decreasing the likelihood that the current study will add to the existing literature base.
Author Response
We thank the reviewer for their time to provide feedback for this manuscript. Your recommended edits and concerns have improved the quality of our work.
This manuscript describes an evaluation of an interprofessional simulation activity between nursing and pharmacy students within the context of a medication dosage form modification. The general topic of interprofessional education is important and of interest, however, there is also a large (and growing) body of literature in this area in this area describing pharmacy and nursing collaborations. It is unclear how this study contributes to the existing literature base and there are also major concerns regarding the methods used. Specific comments are provided:
Introduction
1a) More information needs to be provided regarding the large body of literature addressing interprofessional education efforts between nursing and pharmacy students, including current gaps that this study is intended to address.
Reply: This paper is a communication and so of necessity we have kept the introduction brief. The focus of the paper is the learning opportunity provided by the immersive simulation between nursing and pharmacy students, not the interprofessional aspects. We have emphasised that we have not been able to locate any other papers which use this approach.
1b)Why is oral dosage form modification important to study specifically rather than more general medication activities and responsibilities between nurses and pharmacists? What is not transferable from existing literature that might differ for this specific activity?
Reply: Additional introductory comments made regarding the significance of the issue of Inappropriate dosage modification in both residential care and hospital settings. To our knowledge, there have been no published studies looking at IPE between pharmacy and nursing students involving dosage form modifications in an immersive simulation environment.
Methods
2) There is no study design identified.
Reply: This is a mixed methods pilot study incorporating survey and reflection. This has been added to the abstract lines and methods.
3) Line 73: How was face validity established? It is not enough to say it was established.
Reply: Face validity was established through reviews by both nursing and pharmacy academic staff.
4) Consider placing the case in a table instead of in-text.
Reply: This has been completed and the case is now included in Table 1. An overview of the other three cases utilised in this study is provided in the methods.
5) Lines 109-111: Why did the authors choose to create their own survey rather than use an existing (and potentially validated survey) that measured the same outcomes? Why was it important to ask questions about if students felt the activity was "interesting"? What was the utility of the item asking if this was a "great experience"?
Reply: We thank reviewer 2 for their comment. One of our intentions is to establish students’ interest and engagement with an IPE immersive simulation exercise, as this was a pilot. Enthusiasm and excitement are key motivators for learning. We appreciate that a validated survey will need to be used in future studies.
6) Lines 118-119: Do the authors mean that the responses were coded using the identified categories? What was the outcome of this analysis (i.e. what were done with the codes)?
Reply: Inductive content analysis was performed as described in the methods.
Results
7) Lines 133-134: It would be helpful to identify the number of students who provided open-ended responses.
Reply: 16 nursing students and 8 pharmacy students provided open-ended responses.
8) Lines 135-149: Just listing quotes is not necessarily meaningful without some additional context to them. For example, you identify themes and associated quotes (which is a standard way to present this data), but these quotes simply stand alone with no explanation.
Reply: There were only limited quotes, all were short and therefore, these have not been thematically analyzed. We have presented a limited number of quotes, three from nursing students and four from pharmacy students, which represented the students’ thoughts immediately after the completion of the session.
9) Themes are presented, but there was no description in the Methods of how you got from categories and codes to themes. If themes were developed, please discuss any qualitative validation steps taken.
Reply: These details have been included in the methods section.
Discussion
10) Line 186: It was not clear in the Methods that teaching was part of the simulation activity. Please provide more detail in the Methods about the responsibilities of each profession during the simulation.
Reply: We have altered the word teach in the discussion to “provide advice and guidance”. The role of the pharmacy students is outlined in the example scenario.
Lines 18-20
Fourteen pharmacy students were also present in the simulated health environment, observing the work of the nursing students and available to provide advice regarding medication administration to the nursing students.
Providing advice is also emphasised in the case study
11) Lines 190-198: If this is a major takeaway from the study, it does not add to the existing literature base and is not specific to the oral dosage form modification activity that supposedly is the unique contributor of this study.
Reply: There is no formal assessment to evaluate how this exercise improves students’ knowledge on oral dosage form modification in this pilot study. Dosage form medication is routinely performed in clinical practices, and it is an advanced field of practice. Our intention is to introduce this topic for nursing and pharmacy students to provide them with dosage form modification experience in an immersive simulation environment, which mimics interprofessional collaboration in the real world. This is important to prepare them with skills and knowledge to advise/perform dosage form modification in their practices. Future work will focus on introducing pre- and post-exercise assessments to examine their knowledge and skills in this area.
12) Lines 199-206: Other interprofessional education studies have addressed these limitations or did not have these limitations present, decreasing the likelihood that the current study will add to the existing literature base.
Reply: We thank the reviewer for their comment. Despite only a small number of pharmacy students who participated in this study, as it is designed as a pilot study, the outcome of this study is positive and has led to an integration of this IPE activity into both nursing and pharmacy curricula from 2023 onwards. To our best knowledge, there is no report in the literature on study design that utilised an immersive simulation environment to introduce IPE on dosage form modification. Therefore, this study helps to inform on the utility and feasibility of employing this pedagogical method to deliver IPE.
Reviewer 3 Report
The abstract could be strengthened in the following ways:
1) "student nurses" should be changed to nursing students - in the one sentence where "student nurses" would now appear twices, the final "student nurses" could be replaced with "them".
2) Please add the number of nursing and pharmacy students participating in this pilot project.
3) It would be helpful to identify the type of research conducted - is it descriptive survey research with a qualitative component?
Method section: Please identify here as well the type of research conducted.
Author Response
We thank the reviewer for their feedback.
The abstract could be strengthened in the following ways:
1) "student nurses" should be changed to nursing students - in the one sentence where "student nurses" would now appear twice, the final "student nurses" could be replaced with "them".
Reply: “Student nurses” has been replace with “nursing students” in the abstract to ensure consistency throughout the manuscript.
2) Please add the number of nursing and pharmacy students participating in this pilot project.
Reply: Number of nursing and pharmacy students have been added in the abstract.
3) It would be helpful to identify the type of research conducted - is it descriptive survey research with a qualitative component?
Reply: We thank the reviewer for requesting the type of research conducted to be clearly identified in the abstract and methods. This is a mixed methods pilot study incorporating survey and reflection.
4) Method section: Please identify here as well the type of research conducted.
Reply: The type of research conducted has been added to the methods section.